# Cellular Localization of Two *Rickettsia* Symbionts in the Digestive System and within the Ovaries of the Mirid Bug, *Macrolophous pygmaeus*

**DOI:** 10.3390/insects11080530

**Published:** 2020-08-13

**Authors:** Maria Dally, Maya Lalzar, Eduard Belausov, Yuval Gottlieb, Moshe Coll, Einat Zchori-Fein

**Affiliations:** 1Department of Entomology, RH Smith Faculty of Agriculture, Food and Environment, The Hebrew University of Jerusalem, POB 12, Rehovot 7610001, Israel; maria.dally@mail.huji.ac.il (M.D.); moshe.coll@mail.huji.ac.il (M.C.); 2Department of Entomology, Newe-Ya’ar Research Center, ARO, Ramat-Yishay 30095, Israel; 3Bioinformatics Service Unit, University of Haifa, Haifa 3498838, Israel; maya.lalzar@gmail.com; 4The Institute of Plant Sciences, The Volcani Center, ARO, HaMaccabim Road, Rishon LeZion 7528809, Israel; eddy@agri.gov.il; 5Koret School of Veterinary Medicine, The Hebrew University of Jerusalem, POB 12, Rehovot 76100, Israel; gottlieb.yuval@mail.huji.ac.il

**Keywords:** bacteriocyte, bacterial microbiome, FISH, omnivory

## Abstract

**Simple Summary:**

Like most insects, those that feed on both prey and plant materials harbor symbiotic bacteria in their body. Yet the involvement of bacteria in the feeding habits of these omnivorous consumers has yet to be investigated. In the present study, we took the first step toward testing the hypothesis that bacterial symbionts are involved in the feeding habits of the omnivorous bug *Macrolophus pygmaeus*. We (I) characterized the microbiome (the assembly of bacteria and fungi) of *M. pygmaeus*, and (II) determined the identity and location of the most dominant bacteria species within the host body. We found that *M. pygmaeus* microbiome is dominated by two *Rickettsia* species, *R. belli* and *R. limoniae*. These bacteria are found in high numbers in the digestive system of the bug, each exhibiting a unique distribution pattern, and for the most part, do not share the same cells in the gut. These results strongly suggest that the host bug may gain some nutritional benefits by hosting the two dominant symbiotic bacteria in its gut.

**Abstract:**

Bacterial symbionts in arthropods are common, vary in their effects, and can dramatically influence the outcome of biological control efforts. *Macrolophus pygmaeus* (Heteroptera: Miridae), a key component of biological control programs, is mainly predaceous but may also display phytophagy. *M. pygmaeus* hosts symbiotic *Wolbachia*, which induce cytoplasmic incompatibility, and two *Rickettsia* species, *R. bellii* and *R. limoniae*, which are found in all individuals tested. To test possible involvement of the two *Rickettsia* species in the feeding habits of *M. pygmaeus*, we first showed that the microbiome of the insect is dominated by these three symbionts, and later described the distribution pattern of the two *Rickettsia* species in its digestive system. Although both *Rickettsia* species were located in certain gut bacteriocyes, in caeca and in Malpighian tubules of both sexes, each species has a unique cellular occupancy pattern and specific distribution along digestive system compartments. Infrequently, both species were found in a cell. In females, both *Rickettsia* species were detected in the germarium, the apical end of the ovarioles within the ovaries, but not in oocytes. Although the cause for these *Rickettsia* distribution patterns is yet unknown, it is likely linked to host nutrition while feeding on prey or plants.

## 1. Introduction

Colonization by microorganisms is a universal phenomenon among animals including insects, one of the most successful groups in the animal kingdom [1]. Various types of symbiotic associations with microorganisms including fungi, viruses, protozoans, and particularly bacteria, have contributed to this success. Indeed, bacterial associations are widespread throughout the Insecta [2,3]. In recent decades, the working definition of a symbiont has been expanded to include all microbes that colonize animals, making up the host microbiome [4]. The insect microbiome is a dynamic microbial community which shapes many life history traits of its host [5]. Some symbiotic bacteria, for example, induce sex ratio distortion through mechanisms such as male killing, parthenogenesis, feminization, or cytoplasmic incompatibility [6]. Since microbiota account for up to 1–10% of the biomass of insects [3], they have become the focus of intense research.

Bacterial symbionts can generally be divided into two main categories: inherited intracellular symbionts, which are vertically transmitted from one host generation to the next, and extracellular symbionts, which are acquired from the environment [7]. Symbionts can also be categorized as either obligatory for the survival and development of their hosts, in that they provide digestive capabilities or supplement the hosts’ diet with essential nutrients, or facultative, influencing host fitness by altering host traits and capabilities [8].

It has been suggested that symbionts may be involved in determining the range of food sources that the host is able to exploit [9]. For instance, in the facultative seed-eating predatory beetle *Harpalus pensylvanicus* (Carabidae), more seeds from *Chenopodium album* were consumed in the presence of the symbiont *Enterococcus faecalis* than in its absence [10]. Other studies present evidence that symbionts may provide nutritional benefits to their hosts by modifying plant defensive pathways, thus enabling the insects to overcome plant defenses, as was found for the chrysomelid beetles *Diabrotica virgifera* on maize [11] and *Leptinotarsa decemlineata* on *Solanum lycopersicum* [12].

Insects of the suborder Heteroptera, the true bugs, comprise around 40,000 species worldwide, some of which are important predators of a wide range of phytophagous pest insects [8,13], and thus serve in biological control programs. Some others are important agricultural pests [14]. It has been known for over 50 years that heteropterans harbor symbiotic microorganisms; in many species, the ability to survive and reproduce on plant materials, prey items, or both resources in various mixtures has been attributed to the presence of symbiotic bacteria. Previous work suggests that the absence of symbionts of the genus *Burkholderia* in the broad-headed bugs *Riptortus clavatus* is responsible for a decrease in host fitness [15].

Within the Heteroptera, the nearly 10,000 species of the family Miridae have a diet breadth that includes various levels of specialization for herbivory, carnivory, or both (i.e., omnivory) [16]. Commercially used mirids, including *Macrolophus pygmaeus* and *Nesidiocoris tenuis*, are natural enemies of pestiferous arthropods and, as such, serve as a component of integrated pest management and biological control programs [17]. *Macrolophus pygmaeus* may also display phytophagous habits; it has been shown to successfully develop and oviposit, although at a lower rate, when feeding exclusively on leaves of tomato or eggplant [18]. In contrast, *Nesidiocoris tenuis* is able to feed on plants, but cannot complete its development in the absence of prey [19]. Although the involvement of bacteria in the omnivorous feeding habits of mirid bugs has yet to be investigated, some evidence supports this possibility. The omnivorous bug *N. tenuis*, for example, is known to harbor both *Rickettsia* and *Wolbachia* endosymbionts, and both have been found not only in host ovaries, but in the gut tissue as well [20]. Bacteria of these same genera have been reported by Machtelinckx et al. [21] in *M. pygmaeus*, a key natural enemy of various economically important agricultural pests of greenhouse vegetable crops [17]. Two *Rickettsia* species, *R. bellii* and *R. limoniae*, were identified by Machtelinckx et al. [21]. The commercial application of *M. pygmaeus* as a biological control agent has been limited by its potential to inflict significant damage by feeding on crop plants when prey becomes scarce [22]. Therefore, the overall objective of our ongoing research is to test the possible involvement of bacterial symbionts in the omnivorous feeding habits of *M. pygmaeus*. To this end, we (I) characterized the microbiome of *M. pygmaeus*, and (II) determined the identity and location of two *Rickettsia* species within its body. The results obtained are reported here.

## 2. Material and Methods

### 2.1. Insect Rearing

A culture of *M. pygmaeus* was established in February 2018 with 50 adults (30 females and 20 males) obtained from a commercial biological control company (BioBee Sde Eliyahu Ltd.). The bugs (Figure 1) were kept in ventilated cages in a climate chamber (25 ± 1 °C 16:8 L:D) and offered symbiont-free frozen *Ceratitis capitata* eggs as prey, and tomato seedlings (cv. “Beefsteak”, M. Ben-Shahar Ltd., Tel Aviv, Israel) as both food and oviposition substrate. The seedlings were replenished twice weekly.

### 2.2. Microbiota Characterization

To identify the bacterial and fungal community in *M. pygmaeus*, total genomic DNA was extracted using SDS buffer, as described in [23], from five replicates, each consisting of 20 adult females. The DNA was used as a template for PCR amplification of the V4–V5 variable regions of microbial 16S ribosomal RNA gene [24] using primers (forward 515F and reverse 926R) and the V7–V8 regions of the fungal 18S ribosomal RNA gene using primers FF390 and FR1 [25]. A two-stage targeted amplicon sequencing protocol was used as previously described [26]. Libraries were loaded onto a MiSeq v3 flow cell and sequenced (2 × 300 paired end reads) using an Illumina MiSeq sequencer. PCR amplifications, library preparation, and sequencing were performed at the University of Illinois at Chicago Sequencing Core. Raw sequence data was processed with the Dada2 pipeline [27] in R using package ‘dada2′ (version 1.14.0). Fastq formatted reads were trimmed and filtered for low quality using the command ‘filterAndTrim’ with parameters maxEE = 2, maxN = 0, trimleft = 15. Error rate estimation was carried out using ‘learnerror’ command with default parameters but the randomize parameter set to TRUE, in order to sample nucleotides and reads for model building randomly across all samples. The dada2 algorithm was subsequently implemented for error correction and a count table containing the amplicon sequence variants (ASVs), forward and reverse reads were merged using the ‘mergePairs’ command and counts per sample was produced. Suspected chimera was detected and filtered out using the command ‘removeBimeraDenovo’ using default parameters. For each ASV, taxonomy was inferred by alignment to the Silva non-redundant small subunit ribosomal RNA database (version 132, Ref. [28]) using command ‘assignTaxonomy’ with default parameters but setting minimum bootstrap confidence value to 80%. For the bacterial dataset, non-bacterial ASVs, such as of mitochondrial, or chloroplast origin or unclassified ones, were removed. The resulting count matrix for each ASV in each sample and their taxonomic annotations were used for further analysis.

### 2.3. Screening for Rickettsia bellii and Rickettsia limoniae Prevalance

To study the frequency of *R. bellii* and *R. limoniae* in *M. pygmaeus*, the symbionts were screened using PCR. Fourteen *M. pygmaeus* adults, seven females and seven males, were retrieved from the lab colony and placed in 70% ethanol until processing. Individual insects were ground separately in 90 µL lysis buffer [29] and checked by PCR with species-specific primers for the 16S rRNA gene of *R. bellii* and *R. limoniae* (Table 1). Negative controls contained sterilized water and DNA of *Rickettsia*-free sweet potato whiteflies (*Bemisia tabaci*). DNA of whiteflies harboring the bacterium served as a positive control for *R. bellii*, while the positive control for *R. limoniae* was the DNA used for deep sequencing, in which the bacterium has been identified. PCR procedures for adults were conducted in a final volume of 25 μL containing 10 uL G013-dye 2X PCR Taq MasterMix-abm, 10 pmol per microliter of each primer, 10 μL DDW and 3 μL of DNA. PCR was carried out under the following conditions; *R. bellii*: 2 min at 94 °C, 36 cycles of 30 s at 92 °C, 30 s at 60 °C, 30 s at 72 °C, and a final extension of 5 min at 72 °C. *R. limoniae*: 2 min at 94 °C, 36 cycles of 30 s at 92 °C, 30 s at 55 °C, 30 s at 72 °C, and a final extension of 5 min at 72 °C. PCR products were stained with fluorescent dye (8 μL SafeView™ nuclear stain), electrophoresed on 1.5% agarose gels × TAE-buffer, stained with ethidium bromide, and visualized under UV-light (Bio-Rad Gel Doc XR System, 254 nm).

### 2.4. Location of R. bellii and R. limoniae in M. pygmaeus

#### 2.4.1. Morphology of the Digestive System

To characterize the morphology of female and male digestive systems, over 50 adult females and 50 males of *M. pygmaeus* were dissected on microscope slides in physiological saline, using fine needles. Ovaries and guts were dissected under a stereoscope (Nikon SMZ-1000) by removing the head, anchoring the body, and gently pulling the edge of the abdomen until the ovaries and digestive system were revealed. They were photographed under a 3D digital microscope (Hirox RH-2000, Jyfel, Bâtiment A F-69760, Limonest, France).

#### 2.4.2. PCR for Ovaries and Digestive System

To detect the presence of *R. limoniae* and *R. bellii*, nine females and nine males were dissected individually, as described above. PCR were performed separately for dissected ovaries and the digestive system, as described above, with the following alteration: ovaries and the digestive system were ground separately in 40 µL of Lysis buffer. PCR were performed in 13 μL final volume containing 5 μL G013-dye 2X PCR Taq MasterMix-abm, 10 pmol per microliter of each primer, 5 μL DDW, and 2 μL of DNA.

#### 2.4.3. Fluorescence in Situ Hybridization (FISH)

To determine the location of the two dominant bacteria, *R. limoniae* and *R. bellii*, in the reproductive organs and digestive system, fluorescent in situ hybridization (FISH) was performed. The analysis was carried out following the protocol described by Gottlieb et al. [30] for whole mounted samples, with slight modifications. Each of over 50 ovaries and 50 digestive tracts were placed individually in a drop of 1X PBS under a stereomicroscope, fixed for 5 min in Carnoy’s fixative (chloroform: ethanol: glacial acetic acid, 6:3:1), then washed twice in hybridization buffer (20 Mm TRIS-HCL pH 8.0; 0.9 M NaCl; 0.01% SDS; 30% Formamide [F9037] (Sigma-Aldrich Products Ltd., Rehovot, Israel) without probe, and then hybridized overnight in hybridization buffer with fluorescent probe at 25 °C. Two DNA probes used for the analysis were the *R. bellii*-specific probe CY3 and *R. limoniae*-specific probe CY5 (Table 1). A no-probe experiment and the hybridization of a digestive system were used as controls. Images were acquired using an OLYMPUS IX 81 (Japan) inverted laser scanning confocal microscope (FLUOVIEW 500) equipped with 405, 561, 640 nm laser lines, a UplanApo 10 ×/0.4 NA dry objective, and PlanApo 40 ×/0.9 NA and 60 ×/1.0 NA water immersion objectives. For DAPI; 4′, 6-diamidino-2-phenylindole is a fluorescent stain which binds preferentially to the AT-rich regions of dsDNA, 405  nm excitation light and a BA430-460 nm barrier filter were used; for the detection of fluorochrome CY3, 561 nm excitation light and a BA575-620 nm barrier filter were used. For the detection of fluorochrome CY5, a 640 nm laser line and a BA 660IF filter were used. When DAPI, CY3 and CY5 were detected in a single sample, a dichroic mirror 405/488/561/640 was used. In all cases where more than one dye was monitored, sequential acquisition was performed. Confocal optical sections were obtained at increments of 5 µm, 1.3 µm and 0.8 µm for ×10, ×40 and ×60 objectives, respectively.

## 3. Results

### 3.1. Microbiota Characterization

The bacterial data set originating from five samples (pools of 20 female bugs each), resulted in a total of 94,293 reads mapped to 28 ASVs which were further analyzed (Appendix A). Three ASVs, two *Rickettsia* populations and one *Wolbachia* population dominated the *M. pygmaeus* microbiome with cumulative relative abundance values of 94.8–98.7% among the five replicates. The similarity between two *Rickettsia* ASV was 96% (390 of 407 nucleotides of pairwise aligned sequences), indicating that two separate species of *Rickettsia*, namely *R. limoniae* and *R. bellii*, occur in the studied *M. pygmaeus* population. Only five other ASVs were prevalent in all five samples and included different Actinobacteria. The remaining 20 ASVs included various Actinobacteria, Alphaproteobacteria, and Gammaproteobacteria, with prevalence of 2/5 (1 ASV) and 1/5 (19 ASVs) only (Figure 2 and Appendix A). In the fungal dataset, most sequences were of host origin (94.4–98.0%). The remaining sequences were assigned to 21 fungal ASVs, the majority of which belonged to Ascomycota (Appendix A). Due to the low quantity of sequence data, fungi were not further analyzed. Moreover, the PCR performed with species-specific primers showed that all females and males of *M. pygmaeus* originating from the lab population harbored the two *Rickettsia* species.

### 3.2. Location of R. bellii and R. limoniae in M. pygmaeus Body

#### 3.2.1. Morphology of Digestive System

Microscopic observations revealed the structural configuration of the ovaries (Figure 3A) and female and male guts (Figure 3B,C). While the female digestive system was longer and wider than that of the males, the alimentary tract of both sexes displayed similar division into morphologically distinct regions: the foregut, a tubular region directly connected to mouth; the first, anterior region of the midgut, which was large and sac-like; the second, tubular region of the midgut; the third midgut region, which was soft and moderately swollen; and the fourth, posterior, moderately swollen, region of the midgut that was connected to the hindgut at the joining site of the Malpighian tubules.

Two specific outgrowth caeca which differ between females and males appear in the tissue of the third midgut region: in males, they appeared larger, abutting the digestive system, and were located at the posterior end of the third midgut region. In contrast, in females, the structures were smaller, not in direct full contact with the digestive system, and located at the posterior section of the third midgut region, but not at its end (Figure 3B,C).

#### 3.2.2. PCR for Ovaries and Digestive System

A diagnostic PCR using *Rickettsia*-specific primers on eighteen female and male digestive systems and ovaries showed that all *M. pygmaeus* individuals tested harbored both *Rickettsia* species.

#### 3.2.3. Fluorescence in Situ Hybridization

In situ hybridization targeting bacterial 16S rRNA confirmed the location of the two *Rickettsia* species within the ovaries and the digestive system. In the ovaries, *R. bellii* and *R. limoniae* were concentrated in the germarium, at the apical end of the ovarioles (Figure 4A). *R. limoniae* was scattered throughout the germarium, whereas *R. bellii* presented mainly in clusters (Figure 4B).

FISH analysis revealed the presence of large numbers of the two *Rickettsia* species throughout the female and male digestive systems. In most cases, they were located in different specific host cells (hereafter, bacteriocytes) (Figure 5A and Figure 6A). Infrequently, we found the two *Rickettsia* species sharing the same bacteriocyte (Figure 7 and Figure 8). The bacteriocytes for *R. limoniae* were distributed throughout the entire digestive system, while *R. bellii* was located primarily in the foregut and midgut. FISH targeting bacterial 16S rRNA visualized the *R. limoniae* and *R. bellii* bacteriocytes in the inner of the two outgrowth caeca regions, in the tissue of the third midgut section of females and males (Figure 5B and Figure 6C). Furthermore, FISH analysis detected *R. bellii* and *R. limoniae* within the Malpighian tubules of females and males; *R. limoniae* was observed in all examined Malpighian tubules, whereas *R. bellii* was not always present (Figure 5C).

## 4. Discussion

Bioinformatic analysis in this study revealed that the microbial community associated with a laboratory strain of *Macrolophus pygmaeus* is composed of three dominant endosymbionts: *Wolbachia* sp., *Rickettsia limoniae* and *R. bellii*. An earlier phylogenetic analysis similarly demonstrated that the two *Rickettsia* species in *M. pygmaeus* are related to two different clades [21]. One falls within the ‘Bellii’ group, together with *Rickettsia* reported from several agricultural pests, such as the two-spotted spider mite (*Tetranychus urticae*), the pea aphid (*Acyrthosiphon pisum*), and the sweet potato whitefly (*Bemisia tabaci*). The second *Rickettsia* belongs to the ‘limoniae’ group, which has been reported from the microbiome of non-agricultural insects including the cranefly *Limonia chorea* [31]. The observed similarities in symbionts among arthropod species may suggest that bacteria undergo horizontal transfer between species in the environment. Such a transfer may occur via feeding; omnivorous species such as *M. pygmaeus* may acquire the symbionts directly by consuming infected prey or indirectly by feeding on plants shared with other phytophagous insects. Likewise, other species that feed on the same host plants might subsequently take up micro-organisms transferred by the bug to the plants by way of its piercing-sucking mouthparts. Caspi-Fluger et al. [32] had demonstrated such horizontal transmission of *Rickettsia* between two different whitefly species, through their shared cotton host plants. A similar through-the-plant transmission of intracellular bacteria such as, *Rickettsia*, *Wolbachia*, and *Cardinium,* was reviewed by Chrostek et al. [33] in *Euscelidius variegatus* leafhoppers. In addition, symbionts may be transferred between hosts by their shared parasitoids. For example, *Hamiltonella defensa* and *Regiella insecticola* may be transferred between aphid hosts by parasitoid wasps [34].

Machtelinckx et al. [21] demonstrated the presence of *Wolbachia* and the two *Rickettsia* species in the ovarioles of *M. pygmaeus*, indicating the presence of vertical transmission. Here, we described each *Rickettsia* species and demonstrated that both are found in the germaria at the tip of the ovarioles, but each species had a unique distribution pattern: while *R. limoniae* was scattered throughout the germarium, *R. bellii* appeared mainly in clusters. Although we did not observe *Rickettsia* in the oocytes, the germarium can be an infection zone of *Rickettsia* for further transmission into the oocytes, as described in the bulrush bug, *Chilacis typhe* [35]. It is thus possible that oocyte infection occurs at a later stage of oocyte maturation, before fertilization and deposition. Alternatively, *Rickettsia* had been shown to invade the oocytes of *Bemisia tabaci* whiteflies during early developmental stages, but is mostly excluded from the oocytes when eggs mature [36].

Many members of the suborder Heteroptera have established relationships with microbes, which inhabit specific outgrowths of the midgut caeca. Mirids have been thought to lack these caeca [16,37], but in this study, two specific outgrowths of the caeca were clearly in evidence in the third midgut region. They differed in appearance between males and females. To our knowledge, this is the first report of such structures in the Miridae. A possible nutritional function of these caeca in *M. pygmaeus* is suggested by the presence of bacteriocytes that harbor either *R. bellii* or *R. limonia*, with a few bacteriocytes containing both species.

While *Wolbachia* is known to induce strong cytoplasmic incompatibility in *M. pygmaeus* and is virtually absent from the digestive system [21,38], the role of *Rickettsia* in the biology of this bug is mostly unknown. As the long-term objective of our study is to test the involvement of bacterial symbionts in *M. pygmaeus* diet, we first determined the tissue localization and cellular pattern of the two *Rickettsia* species in the alimentary canal of males and females.

Finding both *Rickettsia* species in *M. pygmaeus* Malpighian tubules may be indicative of a symbiont hosting role of these organs, as was found for nutritional symbionts in ticks [39]. Likewise, Malpighian tubules harbor endosymbionts that appear to contribute to the physiological function of leaf and bark beetles [40]. For instance, *Macroplea appendiculata* and *M. muticare* reed beetles appear to construct underwater cocoons by using the secretion of two bacteria endosymbionts that reside in cells of their Malpighian tubule [41].

Microbes have been reported to be established in both mid- and hind-gut epithelia of arthropods, where the highly convoluted plasma membranes of the cells present a large area for extensive contact with microbial surfaces [37]. Symbionts have in fact been reported in the mid-gut of Heteroptera species belonging to several families, whether in crypts in the posterior region of the mid-gut, in the lumen or on the epithelial walls of the mid-gut itself, or in specialized bacteriomes [42,43,44]. *Rickettsia* were reported in the nuclei and cytoplasm of midgut epithelial cells of the plant bug *Stenotzls birtotatus* [45], in the midgut cells and lumen of *Bemisia tabaci* [30], and in the lumen along the digestive tract of *Nesidiocoris tenuis* [20]. These symbionts appear to enhance the fitness of their heteropteran hosts, may be through nutritional function; the absence of symbionts resulted in retarded growth, increased mortality, and/or sterility [42,46]. Furthermore, Feldhaar et al. [47] demonstrated that *Blochmannia*, an intracellular midgut endosymbiont of *Camponotus floridanus*, provides these ants essential amino acids, and that it may also play a role in nitrogen recycling via its functional urease. Interestingly, removal of all three symbionts from *M. pygmaeus* resulted in higher sensitivity to freezing conditions [48]. It is not yet clear which of the symbionts causes this negative effect by its absence.

FISH results showed the presence of bacteriocyte-like cells within the digestive system of *M. pygmaeus*. Each of the *Rickettsia* species was found to be hosted in a different bacteriocyte within the digestive gut compartments. In rare instances, the two *Rickettsia* species were found to share the same bacteriocyte. The intracellular arena hypothesis, which states that genetic exchange can occur in communities of bacterial endosymbionts that infect the same cellular environment of a shared host, is supported by documented findings from the dipteran *Drosophila simulans* and the hymenopteran *Nasonia vitripennis* [49]. To our knowledge, however, this is the first report of two *Rickettsia* species sharing the same bacteriocyte in insects. The association of different bacterial species together in the same host cell may suggest that each symbiont induces unique effects, and that they may act synergistically [50]. In the context of multiple infections with vertically transmitted symbionts, the bacterial partners are limited to a restricted shared environment and the evolutionary fate of all partners is tightly linked. If the transmission of each partner depends on the transmission of the others, cooperative interactions and communication among symbionts can be expected [50]. However, since the cohabitation of a single bacteriocyte by the two *Rickettsia* species was rare in the studied system, it may indicate antagonistic interactions through competitive displacement [50,51]. Notably, *R. limoniae* was more broadly distributed along the host digestive system than *R. bellii* and appeared in a scattered pattern compared to the clustering of *R. bellii*. This may suggest a different interaction with the host cellular environment and may be the result of different host control mechanisms over its symbionts.

## 5. Conclusions

In conclusion, our results describe the presence of two congeneric *Rickettsia* species in *M. pygmaeus* ovaries and alimentary canal. That unique distribution pattern suggests a possible nutritional benefit conferred by these endosymbionts on their omnivorous host, via either complementary or synergistic interaction.

## Figures and Tables

**Figure 1 insects-11-00530-f001:**
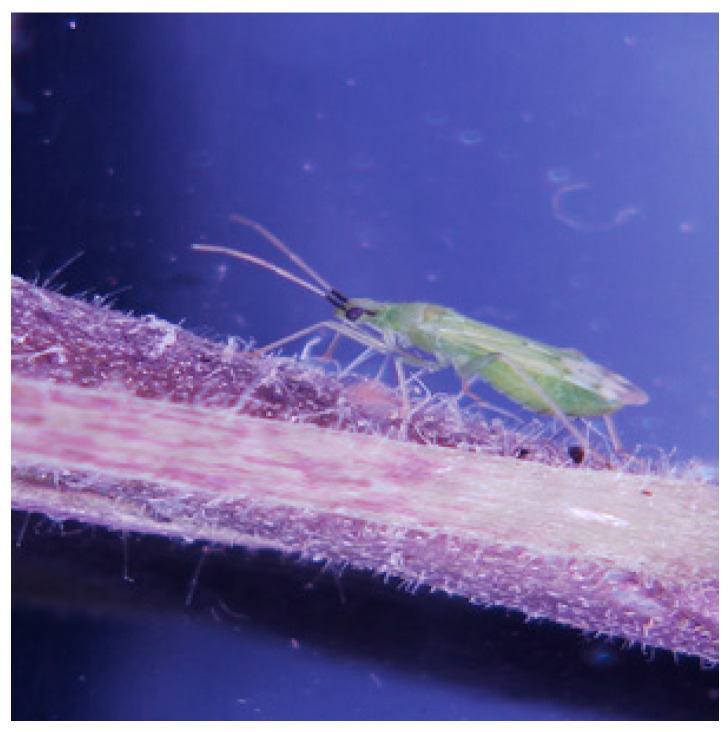
Adult female *Macrolophus pygmaeus*.

**Figure 2 insects-11-00530-f002:**
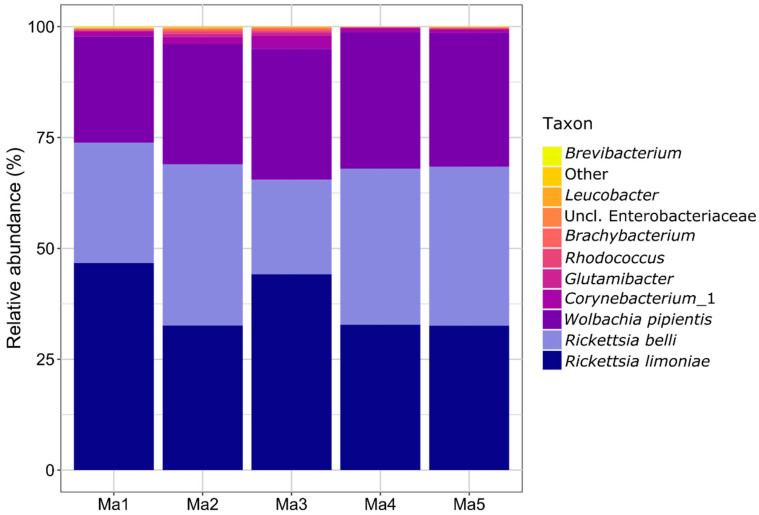
Relative abundance (%) of bacterial microbiota composition of *M. pygmaeus* (five replicates, pools of 20 females each) based on amplicon sequencing of bacterial partial 16S rRNA gene (V4 region). Amplicon sequence variants were determined and quantified per sample using the Dada2 pipeline and taxonomically assigned by alignment to the Silva rRNA gene database (Appendix A). Ma, *M. pygmaeus*.

**Figure 3 insects-11-00530-f003:**
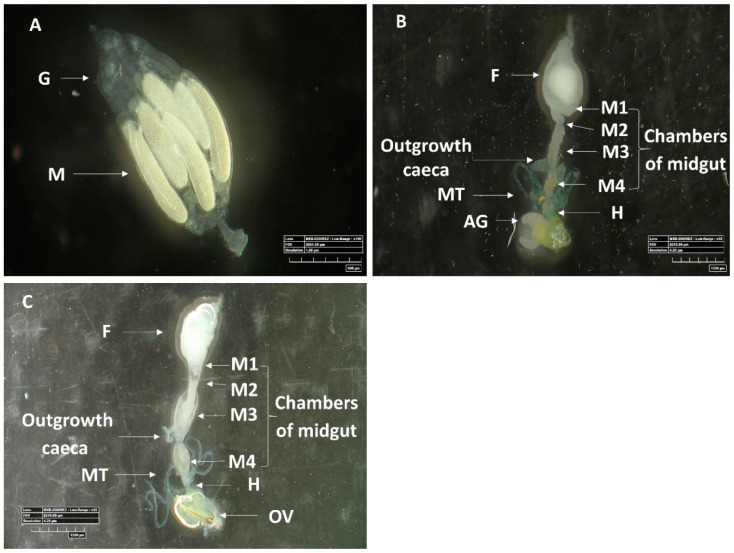
Images of *M. pygmaeus* ovary and male and female digestive systems. (**A**) An isolated ovary with several ovarioles; M, mature oocyte; G, Germarium, (**B**) Isolated male and (**C**) female digestive systems; F, foregut; M1, midgut first section; M2, midgut second section; M3, midgut third section (with outgrowth caeca); M4, midgut fourth section; MT, Malpighian tubules; H, hindgut; AG, accessory gland; OV, ovipositor. Images A-C were acquired by 3D digital microscopy using identical settings.

**Figure 4 insects-11-00530-f004:**
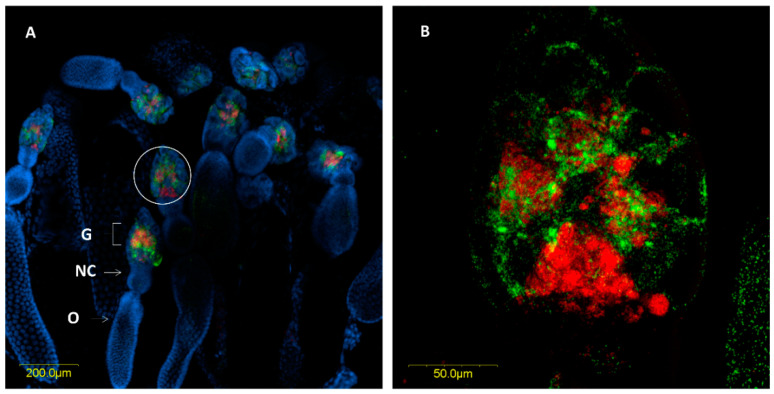
Fluorescence in situ hybridization (FISH) of *M. pygmaeus* ovarioles, *R. bellii* specific probes (red), *R. limoniae* specific probes (green), and DNA dye (blue). (**A**) Ovary with several ovarioles, *R. bellii* and *R. limoniae* are concentrated in the germarium; G, Germarium; NC, nurse cells; O, Oocyte. (**B**) Enlarged section of the germarium (white circle in A). Images A and B represent serial Z sections of 35 μm and 13.6 μm, respectively.

**Figure 5 insects-11-00530-f005:**
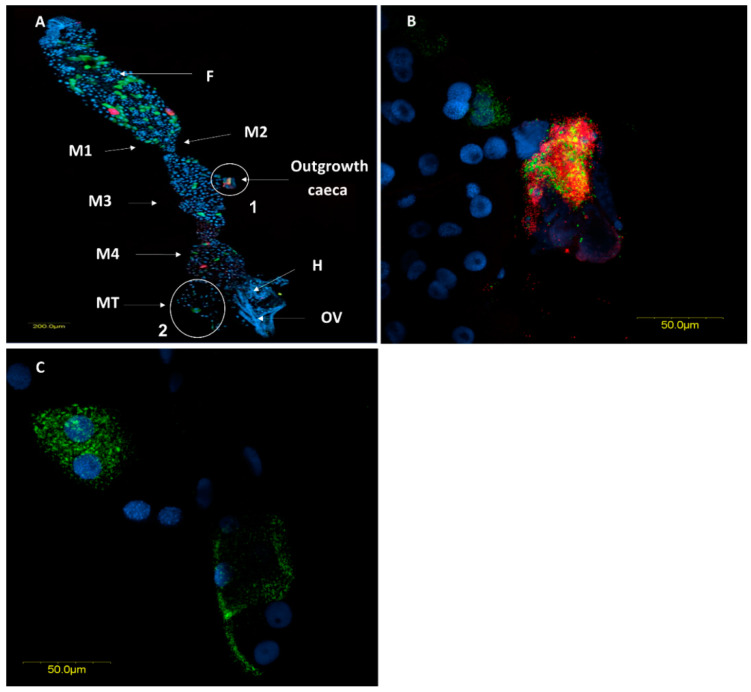
FISH of a female *M. pygmaeus* digestive system (DNA in blue). (**A**) The whole digestive system, *R. bellii* (red), *R. limoniae* (green). Reconstruction: three frames of the same gut; from F to M2 picture number 1, M3 picture number 2, M4 to OV picture number 3. (**B**) Enlarged section of the outgrowth caeca region, in the gut tissue of the third midgut section (white circle 1 in A). (**C**) Enlarged section of Malpighian tubules (white circle 2 in A). Images A number 1, A number 2, A number 3, B and C represent serial Z sections of 20 μm, 55 μm, 50 μm, 12 μm, 17 μm respectively. F, foregut; M1, midgut first section; M2, midgut second section; M3, midgut third section (with outgrowth caeca); M4, midgut fourth section; MT, Malpighian tubules; H, hindgut; OV, ovipositor.

**Figure 6 insects-11-00530-f006:**
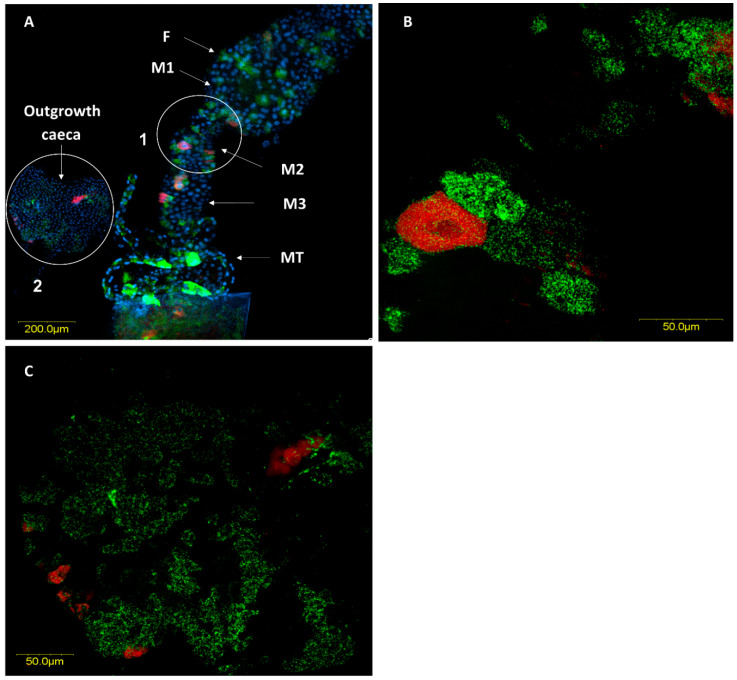
FISH of the male *M. pygmaeus* digestive system (DNA in blue). (**A**) The digestive system, *R. belli*i (red), *R. limoniae* (green), with two outgrowth caeca from the gut tissue of the third midgut section dissected aside (see circle 2). (**B**) Enlarged section from the midgut (see circle 1). (**C**) Enlarged section of the two outgrowth caeca (see circle 2). Images A, B and C represent serial Z sections of 35 μm, 8.8 μm, 37.7 μm respectively. F, foregut; M1, midgut first section; M2, midgut second section; M3, midgut third section; MT, Malpighian tubules.

**Figure 7 insects-11-00530-f007:**
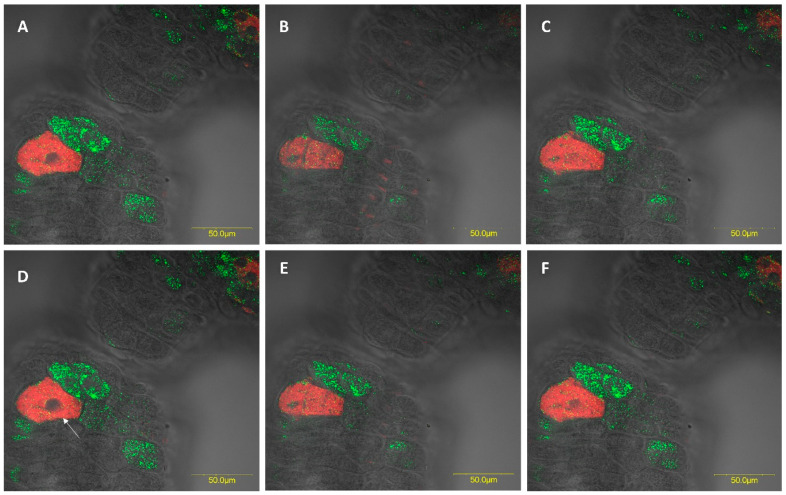
Z sections from part of male midgut of *M. pygmaeus* (circle 1 Figure 6A,B). *Rickettsia bellii* (red) and *R. limoniae* (green) can be seen to inhabit separate bacteriocytes in the gut epithelium, as well as occupying the same bacteriocyte, e.g., in Figure D (see arrow). (**A**–**F**) Serial Z-sections of 8.8 um (0.8 um increasement).

**Figure 8 insects-11-00530-f008:**
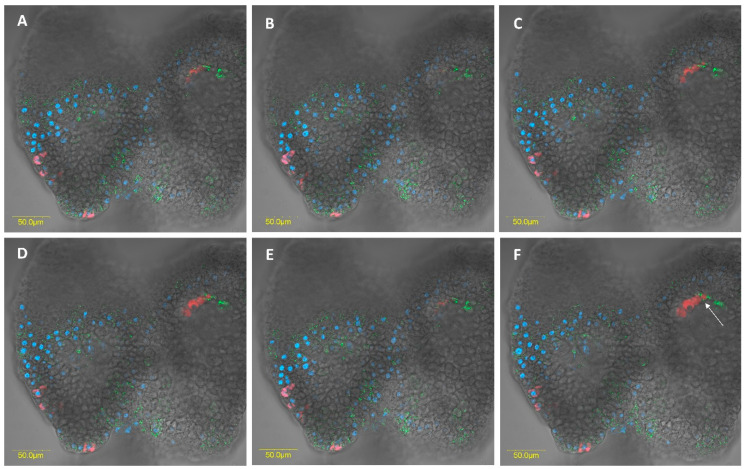
Z sections of two male outgrowth caeca (circle 2 Figure 6C) DNA (blue). *R. bellii* (red), *R. limoniae* (green) can be seen to inhabit separate bacteriocytes, as well as occupy the same bacteriocyte in the outgrowth caeca epithelium, e.g., in Figure F, see arrow. (**A**–**F**) Serial Z-sections of 37.7 um (1.3 um increasement).

**Table 1 insects-11-00530-t001:** Primers and probes used to identify *Rickettsia bellii* and *R. limoniae* in *M. pygmaeus*.

Targeted Gene/Probes	Name	Sequence	Reference
Targeted Genes			
16S rRNA gene of *R. limoniae* and *R. bellii*	Rick-1F	5′-ATACCGAGTGRGTGAYGAAG-3′	[21]
16S rRNA gene of *R. limoniae*	*Ricklimoniae*-F	5′-CGGTACCTGACCAAGAAAGC-3′	[21]
16S rRNA gene of *R. bellii*	*Rickbellii*-R	5′-TCCACGTCGCCGTCTTGC-3′	[30]
16S rRNA *Rickettsia*	1513R	5′-ACGGYTACCTTGTTACGACTT-3′	[30]
Probes			
Rb1-Cy3	*R. bellii*-specific probe	5′-TCCACGTCGCCGTCTTGC-3′	[30]
Rl1-Cy5	*R. limoniae*-specific probe	5′-GCTTTCTTGGTCAGGTACCG-3′	[21]

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
