# Peer review of "Cellular Localization of Two Rickettsia Symbionts in the Digestive System and within the Ovaries of the Mirid Bug, Macrolophous pygmaeus"

_insects, 2020, doi:10.3390/insects11080530_

Round 1

Reviewer 1 Report

This manuscript by Dallt et al. is a nice description of the two Rickettsia species associated with the predatory bug Macrolophus pygmaeus. Although are best known as arthropod-borne pathogens of vertebrates, Rickettsia bacteria are known to form symbiotic associations with several arthropod species but with obscure biological role for their hosts. Here the authors first confirm the results of a previous study (Machtelinckx et al. 2012) that the microbial community of M. Pygmaeus is dominated by 2 bacterial genera (Wolbachia and Rickettsia), and further report on the tropism of the two Rickettsia species in the tissues of M. Pygmaeus. Their findings are interesting especially the different localization patterns observed for the two Rickettsia species and may have implications for the symbiont functional roles in this host system. Bellow are some comments that may help the authors to improve the manuscript.

Line 80: The identity of the 2 Rickettsia species was determined in the same study by Machtelinckx et al. Is not clear which later study the authors refer to.

Line 100: “with minor modifications”. What modifications? DNA isolation protocols may significantly influence 16S amplicon sequencing results. For research reproducibility the modifications of the original protocol should be clearly stated.

Line 121: “prevalence” instead of “abundance”

Line 130: “1 ul of each primer”. Reporting the concentration is more useful.

Line 179: add “was” after ASV

Lines 276-277: Which populations? Is not clear. This sentence seems redundant.

Lines 283-285: You may want to specify or present some particular examples. Are there evidences that Rickettsia bacteria can move horizontaly between host species? See for example Li et al. 2017 on Belii Riclettsia (https://doi.org/10.1093/femsec/fix138) or Pilgrim et al. 2017 on Torix Rickettsia ( https://doi.org/10.1111/1462-2920.13887).

Line 285: “Such transfer may occur via feeding”. Parasitoids can also serve as means for horizontal symbiont transfers.

Lines 285-289: Discussion of relevant literature is missing here. Plant mediated horizontal transmission of insect symbionts including Rickettsia has been reported previously (reviewed here https://doi.org/10.3389/fmicb.2017.02237).

Line 293: Replace “ but each in a unique pattern of distribution” with “but each strain show unique distribution pattern”

Lines 294-297: In Bemisia "Rickettsia invades the oocytes during early developmental stages and resides in follicular cells and cytoplasm; it is mostly excluded when the egg matures; however, some bacterial cells remain in the egg, ensuring their transfer to subsequent generations." (doi: 10.1128/AEM.01184-12). Could M.pygmaeus Rickettsia use a similar strategy for transmission. The fixation of Rickettsia in M.pygmaeus deffinetly suggest efficient maternal transmission.

Lines 308-311: This is mostly a results section.

Line 308: maybe replace “which open” with “located”

Lines 313-314: What physiological functions?

Reviewer 2 Report

In their manuscript “Cellular localization of two Rickettsia symbionts in the digestive system and within the ovaries of a Mirid bug, Macrolophous pygmaeus” the authors describe the bacterial and supposedly non-existant fungal microbiome of a pest control strain of the mirid but M. pygmaeus using amplicon sequencing of a small bacterial subunit gene region for each bacteria and fungi, combined with diagnostic PCR. After finding that the fungal microbiota sequencing approach almost exclusively yielded host gene reads, they concluded the absence of a relevant fungal microbiota. The bacterial microbiota was dominated by three species, Wolbachia, Rickettsia bellii and Rickettsia limoniae, all three of them consistently present in all examined samples, followed by a few less abundant, yet consistent Actinobacteria. They further localized the two Rickettsia symbionts with the gut epithelium as well as reproductive tissue of both female and male bugs. Both were present in the female germarium, R. limoniae in the entire, and R. bellii in the fore and midgut, usually separated into different host cells. The authors highlight that several of these findings match previous descriptions, contributing basically the localization of the dual Rickettsia infection in this study. They further conclude that the co-occurrence of two intracellularly located congeneric species in the same cells in some instances would suggest synergistic effects, as well as a nutritional contribution due to the localization within the gut.

While the results seem to be solid, I would suggest presenting results of Fig 2 per sample, and not via all samples pooled. At least I understand the figure caption the way that 5 individual samples were sequenced.  Further, I can’t fully follow the interpretation. While a synergistic effect can not be excluded, I don’t see any data supporting this, instead of just completely unrelated effects, or just co-localization sometimes by chance.

The assumption of a nutritional function is in my opinion a bit more likely, but could be explained a bit better, e.g. that the occurrence of both gut and ovary localization would rather speak against pathogenic infections, as they weren’t found in the remaining body. The authors mention the possibility of a horizontal acquisition, but could improve their argumentation. In addition, mentioning other gut epithelium localized symbionts with experimentally verified nutritional functions e.g. Blochmannia in carpenter ants, would strengthen this argument.

I would suggest revising Fig. 1, and the arguments on both points a bit, perhaps keep different hypotheses resulting from the co-localization at hand without favoring the synergistic effect, but otherwise would support acceptance of the article.

Minor comments:

  • In which combinations were primers listed in table 1 used?
  • Further, target and same seem to be mixed for the FISH probes in table 1
  • L 313/314 multiple publications by G. Koelsch would be available to support nutritional function of Donaciinae beetles
  • Species names should be italized throughout the manuscript
